# Effects of Sublethal Concentrations of Tetracycline Hydrochloride on the Biological Characteristics and *Wolbachia* Titer in Parthenogenesis *Trichogramma pretiosum*

**DOI:** 10.3390/insects13060559

**Published:** 2022-06-20

**Authors:** Xiaoge Nian, Xiaobing Tao, Zhuangting Xiao, Desen Wang, Yurong He

**Affiliations:** Department of Entomology, College of Plant Protection, South China Agricultural University, Guangzhou 510642, China; nianxiaoge@scau.edu.cn (X.N.); t1304716196@163.com (X.T.); zhuangtingxiao@163.com (Z.X.)

**Keywords:** *T. pretiosum*, *Wolbachia*, tetracycline, sublethal concentrations

## Abstract

**Simple Summary:**

*Trichogramma pretiosum* is an important natural enemy of lepidopteran pests. *Wolbachia* is an intracellular endosymbiont that induces parthenogenesis in the parasitoid *T. pretiosum*. Tetracycline antibiotics are widely used to remove endosymbiont *Wolbachia* from insect hosts. However, the sublethal effects of tetracycline on the development of *T. pretiosum* and the population dynamic of *Wolbachia* in *T. pretiosum* are still unclear. In our present study, after treatment with sublethal concentrations of tetracycline over ten generations, the biological parameters (longevity, parasitized eggs, and fecundity) of treated females and the percentage of female offspring were significantly lower than those in the control. Moreover, the *Wolbachia* titer in females sharply declined after two generations of antibiotic treatments and decreased to a lower level even after ten successive generations of antibiotic treatments. In addition, the control group with a higher *Wolbachia* titer produced more female offspring than the tetracycline treatment groups with a lower *Wolbachia* titer. These results provide new insights into the complex interaction between arthropods and *Wolbachia* to antibiotic stress.

**Abstract:**

*Trichogramma pretiosum* Riley is an important natural enemy and biological control agent of lepidopteran pests. *Wolbachia* is an intracellular endosymbiont that induces parthenogenesis in the parasitoid *T. pretiosum*. In this paper, the sublethal effects of the antibiotic tetracycline hydrochloride on the development and reproduction of *T. pretiosum* were studied. Emerged females were fed with sublethal concentrations (LC_5_, LC_15_, and LC_35_) of tetracycline for ten generations. The biological parameters (longevity, parasitized eggs, and fecundity) of treated females significantly reduced compared with the control Moreover, the percentage of female offspring in the treatments significantly reduced, but the percentage of male offspring significantly increased. In addition, the *Wolbachia* titer sharply reduced after two generations of antibiotic treatments, but it could still be detected even after ten successive generations of antibiotic treatments, which indicated that *Wolbachia* was not completely removed by sublethal concentrations of tetracycline. The control lines with higher *Wolbachia* titers produced more female offspring than the tetracycline treatments with lower *Wolbachia* titers, indicating that the *Wolbachia* titer affected the sex determination of *T. pretiosum*. Our results show that sublethal concentrations of tetracycline had adverse effects on the development of *T. pretiosum*, and *Wolbachia* titers affected the sexual development of *T. pretiosum* eggs.

## 1. Introduction

The *Trichogramma* genus (Hymenoptera: Trichogrammatidae) consists of over 600 species of egg parasitoids. Many of them are widely used in biological control programs against agriculture and forestry pests, including Lepidoptera, Hymenoptera, Coleoptera, Neuroptera, and Diptera [1]. *Trichogramma* species are divided into two groups according to their reproductive mode. Generally, the most common reproductive mode is arrhenotoky, where females originate from fertilized diploid eggs and males originate from unfertilized haploid eggs [2]. Thelytoky, known as thelytokous parthenogenesis, is also a form of parthenogenesis [3]. In this case, thelytoky can be under the control of parthenogenesis-inducing (PI) *Wolbachia,* which can change the gamete duplication of their hosts, resulting in nearly 100% female offspring without mating [4,5,6,7]. However, the frequency of infection with *Wolbachia* in *Trichogramma* wasps is low, and there are 16 thelytokous *Trichogramma* species induced by a natural *Wolbachia* infection [8,9,10]. Compared with bisexual lines, thelytokous *Trichogramma* are usually viewed as superior biological control agents due to the following reasons: (1) they have a potentially higher rate of reproduction because all of their offspring are female; (2) the costs of mass-rearing production are reduced because the host does not need to “waste” resources on males; and (3) the field population of thelytokous *Trichogramma* may be easier to establish because released females can produce generations of female offspring without mating.

*Wolbachia pipientis* are the most abundant endosymbiotic bacteria that are ubiquitous in arthropods and nematodes [7]. Approximately 60% of insect species have been infected by *Wolbachia* around the world [11]. *Wolbachia* is primarily transmitted vertically from mother to offspring via the host egg cytoplasm [12,13]. *Wolbachia* induces various types of reproductive modifications in hosts, including cytoplasmic incompatibility (CI) [14,15], male killing [16], feminization [17], and parthenogenesis induction (PI) [4,18], to increase the reproductive success of infected females. The most common way to study *Wolbachia* functions is to compare the physiological differences between infected and uninfected strains. The naturally *Wolbachia*-infected *Trichogramma bourarachae* had a higher fecundity than their uninfected counterparts [19]. In contrast, the arrhenotokous strains of *Trichogramma deion*, *T. pretiosum*, and *T. cordubensis*, which were obtained by feeding antibiotics to *Wolbachia*-infected strains for several generations, produced more offspring than their thelytokous counterparts [20,21]. *Trichogramma kaykai* females had a shorter longevity and low parasitism and emergence rates when they were infected with *Wolbachia* [22,23]. Moreover, *Wolbachia*-infected *Trichogramma brassicae* would parasitize fewer host eggs compared with its bisexual strain [24]. Ning et al. [25] found that *Wolbachia* was a direct factor that caused the occurrence of intersexes in *T. pretiosum*. However, it was unclear whether *Wolbachia* was completely eliminated because the *Wolbachia* titer was not tested in either female or male offspring in the above studies. In recent years, Real-Time RCR (RT-PCR) and other molecular techniques have been widely applied to detect the endosymbiont *Wolbachia* titer and explore its function. Wang et al. [26] used RT-PCR and fluorescence in situ hybridization to test the *Wolbachia* titer and found that *Wolbachia* was advantageous to *Encarsia formosa* fertility because the higher the *Wolbachia* titer, the higher the proportion of female offspring. Li et al. [27] monitored the population dynamics of *Wolbachia* in *Laodelphax striatellus* during and after successive tetracycline treatments; the results showed that *Wolbachia* could be eliminated within the first two generations, and it was not recovered in the following generations. Taken together, these finding clearly indicate that *Wolabchia* can influence host reproduction, physiology, and gene transcription to increase its own survival.

In the present study, the naturally *Wolbachia*-infected *T. pretiosum* (*T. pre^W+^*) population was used as a study subject. Absolute quantitative PCR (AQ-PCR) was applied to detect the population dynamics of the *Wolbachia* titer in *T. pretiosum* under successive stress treatments of tetracycline. In addition, longevity, parasitized eggs, the fecundity of females, the sex ratio of offspring, the life table parameters (*R_0_*, *r_m_*, *λ*, *T,* and *DT*), and the *Wolbachia* titer were evaluated in every two generations. The objectives of this study were as follows: (1) to investigate whether *Wolbachia* affected the biological characteristics of *T. pre^W+^*, (2) to investigate whether *Wolbachia* could be removed using long-term tetracycline treatments to obtain an uninfected line, (3) and to explore the interaction of the *Wolbachia* titer and the parthenogenesis of *T. pre^W+^*. Our research aimed to provide a new insight into the complex interaction between *Wolbachia* and its host.

## 2. Materials and Methods

### 2.1. Rearing and Maintenance of T. pretiosum

*T. pretiosum* Riley infected with *Wolbachia* were provided by the Pest Biological Control Laboratory of South China Agricultural University. The stock population was reared on the eggs of *Corcyra cephalonica* (Stainton) (Lepidoptera: Pyralidae) in glass culture tubes (2.5 × 7.5 cm: 2.5 cm diameter and 7.5 cm height, plugged with cotton) at 25 ± 1 °C, 70 ± 5% relative humidity (RH), and a 14: 10 h light–dark photoperiod in a growth chamber. To rear *C. cephalonica*, we referred to the method proposed by Wang et al. [28]. The eggs of *C. cephalonica* were collected daily and stored at 4 °C in a refrigerator. The eggs were exposed to UV radiation for 1 h to kill the embryos before parasitoid rearing and experiments.

To obtain virgin *T. pretiosum* adults, parasitized *C. cephalonica* eggs were placed individually in small glass tubes (1.0 × 5.5 cm) before emergence. Several tiny 25% honey–water drops were brushed on the wall of the tube as food for emerging wasps. All experiments were carried out at 25 ± 1 °C and 70 ± 5% RH with a photoperiod of 14: 10 h (L: D) in a growth chamber.

### 2.2. Determination of LC_5_, LC_15_, and LC_35_ of Tetracycline Hydrochloride

The antibiotic used in the experiment was tetracycline hydrochloride (Sigma-Aldrich). Tetracycline was diluted with 25% honey–water into the following five concentrations: 1.5 mg mL^−1^, 2 mg mL^−1^, 3 mg mL^−1^, 4 mg mL^−1^, and 6 mg mL^−1^. Furthermore, 25% honey–water was used as the control treatment. A total of 30 newly emerged *T. pretiosum* females (<6 h old) were tested for each tetracycline concentration. After being fed for 24 h, the number of dead parasitoids was counted, and the concentration responses (LC_5_, LC_15_, and LC_35_) were calculated. Three replicates were carried out for each concentration treatment.

### 2.3. Tetracycline Treatments

Sublethal concentrations of tetracycline were prepared: LC_5_ (25% honey–water with tetracycline of 0.79 mg mL^−1^), LC_15_ (25% honey–water with tetracycline of 1.37 mg mL^−1^), and LC_35_ (25% honey–water with tetracycline of 2.46 mg mL^−1^). Furthermore, 25% honey–water was used as the control. A flowchart of the tetracycline treatments is shown in Figure 1.

Approximately 100 newly emerged females (<6 h old) were placed in a glass tube (2.5 × 7.5 cm) and fed with each tetracycline concentration for 24 h. An egg card with surplus *C. cephalonica* eggs was inserted into each tube to be parasitized for 24 h. Next, the parasitized egg card was labeled and maintained in the small glass tube (1.0 × 5.5 cm) until adult emergence (F_1_ generation). Approximately 100 newly emerged females from the F_1_ generation in each concentration were placed in in a glass tube (2.5 × 7.5 cm) and fed with the same tetracycline concentration for 24 h. Then, one *C. cephalonica* egg card was offered and parasitized for 24 h in each treatment, and the parasitized egg sheets were maintained in a growth chamber until adult emergence (F_2_ generation).

The newly emerged F_2_ females (<6 h old) were divided into three assays.

(1)The first assay was carried out to evaluate the sublethal effects of tetracycline on the parasitoid.

The details are as follows: 30 females from the F_2_ generation in each concentration were placed individually into membrane-forming glass dactylethrae (2.3 × 7.4 cm) with a superfluous *C. cephalonica* eggs card and supplied with 25% honey–water solution. The parasitized egg cards were renewed every 24 h until the female died. All the parasitized egg cards were incubated in a growth chamber until adult emergence. The longevity, parasitism capacity, and the fecundity (number of offspring produced per female) of the females were assessed. The sex ratio (SR) of offspring was also recorded for all the emerged adults. Based on the adults’ antennae characteristics [6,25], SR was calculated using the formulas: SR (% female) = [number of females/(number of females + number of males + number of intersex)] × 100; SR (% male) = [number of males/(number of females + number of males + number of intersex)] × 100; SR (% intersex) = [number of intersex/(number of females + number of males + number of intersex)] × 100.

(2)The second assay was conducted to detect the *Wolbachia* titer using AQ-RCR (see quantification of *Wolbachia* below for details).(3)The third assay was conducted to keep the F_2_ generation in order to obtain the F_3_ generation.

Approximately 100 newly emerged females from the F_2_ generation in each concentration were placed in in a glass tube (2.5 × 7.5 cm) and fed with the same tetracycline concentration for 24 h. Then, one *C. cephalonica* egg card was offered and parasitized for 24 h in each treatment, and the parasitized egg sheets were maintained until adult emergence (F_3_ generation). Moreover, the females of the previous generation were treated with same tetracycline concentration for 24 h to obtain the next generation, which was repeated up to the F_10_ generation. The longevity, number of parasitized eggs, the fecundity of females, SR (% female, % male, and % intersex), and the *Wolbachia* titer in were evaluated every two generations after treatment.

### 2.4. Quantifications of Wolbachia

Total DNA extraction was performed in groups containing the same amount of females to reduce the possible variations in the efficiency of the extraction method. gDNA was extracted from *T. pretiosum* adults (n = 10 adults/concentration/generation/biological replicate) using a TIANamp Genomic DNA Kit (Tiangen Biotech; Beijing, China) according to the manufacturer’s instructions. The special primers for the *Wolbachia* surface protein (*Wsp*) were the forward primer (*Wsp*-F) 5′-TGGTCCAATAAGTGATGAAGAAAC-3′ and the *Wsp* reverse primer (*Wsp*-R) 5′-AAAAATTAAACGCTACTCCA-3′. PCR amplification consisted of 12.5 μL 2 × TaqPCRMasterMix, 1μL 10 mM *Wsp*-F, 1 μL 10 mM *Wsp*-R, 1 μL DNA template, and 9.5 μL ddH_2_O in a final volume of 25 μL. The PCR condition was as follows: 95 °C for 3 min (1 cycle); 95 °C for 30 s, 55 °C for 40 s, and 72 °C for 1 min (35 cycles); and 72 °C for 10 min (final extension). gDNA concentration and quality were assessed using a Nanodrop 2000/2001 spectrophotometer (Thermo Scientific, Wilmington, DE, United States) and by 1% agarose gel electrophoresis using Tris-acetate-EDTA (TAE) buffer at a constant voltage (80 V). The target gene from each symbiont was amplified, and the obtained PCR product was purified. The purified products were inserted into the pClone007 blunt vector (Tsingke Biotech, Beijing, China), and they were transformed into TransB (DE3) (TransGen Biotech, Beijing, China) chemically competent cells, which were grown in a Luria–Bertani (LB) culture medium supplemented with 100 μg mL^−1^ ampicillin. Suitable colonies were isolated, cultivated in LB liquid medium, and subjected to plasmid extraction using a high-purity plasmid DNA extraction kit according to the manual protocol (Tsingke Biotech, Beijing, China). The obtained plasmids were subjected to PCR amplification, followed by verification of the insert size on a 1% agarose gel electrophoresis as described earlier. Plasmids containing the correct insert size were used to produce a dilution standard curve for detected *Wolbachia* titers.

The number of copies (N) of the target genes per microliter was determined using the following equation:$$N=\frac{X\;g/\mathrm{uL}\;\mathrm{DNA}\;}{\mathrm{size}\;\mathrm{of}\;\mathrm{the}\;\mathrm{clone}\;\mathrm{in}\;\mathrm{bp}\;x\;660\;}\;6.023\;\times \;10^{23}$$

(X is the Quantity of DNA in g μL^−1^. The fragment size of the clone is equal to the plasmid plus the inserted. 660 g mol^−1^ is the average number weight of 1 DNA bp. 6.023 × 10^23^ is the number of elementary entities in 1 mol (Avogadro constant).

AQ-PCR was used to measure the *Wolbachia* titer. The symbiont titer was obtained as a measure of the number of copies of the selected target gene using PCR-CFX 96 (Bio-Rad, Hercules, CA, United States). The 20 μL AQ-PCR reaction system consisted of 10 μL SuperReal PreMix Plus (2 ×) (Tiangen Biotech; Beijing, China), 1 μL 10 mM forward primer (5′-AGTGGTTGAAGATATGCC-3′), 1 μL 10 mM reverse primer (5′-AGAGTTTGATTTCTGGGG-3′), 1 μL DNA, and 7 μL ddH_2_O. The AQ-PCR condition was 95 °C for 5 min, followed by 40 cycles of 95 °C for 10 s, 64 °C for 20 s, and 72 °C for 25 s, and then 95 °C for 15 s, 60 °C for 1 min, and 95 °C for 15 s. This was repeated three times for each sample, and the whole experiment was repeated three times.

### 2.5. Statistical Analysis

All the data were checked for normality and homogeneity of variances before analysis. The *Wsp* gene copy number per sample was considered as the *Wolbachia* titer. The toxicity of tetracycline and the *Wolbachia* titer were analyzed using one-way analysis of variance (ANOVA). The data on longevity, parasitized eggs, the fecundity of the females, and sex ratio were analyzed using two-way analysis of variance (ANOVA), with tetracycline concentrations and generations as the factors in the analysis. Tukey’s Honestly Significant Difference (HSD) test was used to determine the statistically significant differences between means. All statistical analyses were performed using SPSS software (version 22.0) [29].

Daily schedules of mortality and fecundity were integrated into a life table format [30] and used to calculate life table parameters: probability of an individual surviving to age *x* (*l_x_*), a female at age *x* (*m_x_*), net reproduction rate (*R_0_* = ∑*l_x_m_x_*), mean generation time (*T* = ∑*xl_x_m_x_*/∑*l_x_m_x_*), intrinsic rate of natural increase (*r_m_* = ln*R_0_*/*T*), finite rate of increase (*λ* = *e^rm^*), and doubling time (*DT* = ln*2/r_m_*).

## 3. Results

### 3.1. Concentration–Mortality Response

Concentration–mortality data from the bioassays of the tetracycline treatments showed a good regression fit to the probit model for *T. pretiosum* adults. The slope of the regression line indicated how fast mortality occurred as the concentration increased. The sublethal concentration LC_5_, LC_15_, and LC_35_ values of tetracycline were 0.79 mg mL^−1^, 1.37 mg mL^−1^, and 2.46 mg mL^−1^, respectively (Table 1).

### 3.2. Sublethal Effects of Tetracycline on Biological Parameters of Treated Female Parasitoids and Their Offspring

The dynamics of female longevity, the number of parasitized eggs, fecundity, and the sex ratio (% female, % male and % intersex) of offspring were investigated under successive stress treatments of sublethal concentrations (LC_5_, LC_15_, and LC_35_) of tetracycline in every two generations (Figure 2).

Both tetracycline concentrations (*F* = 84.20; *df* = 3, 621; *p* < 0.001) and generations (*F* = 25.57; *df* = 4, 621; *p* < 0.001) significantly impacted the average longevity of *T. pretiosum* females, as did the interaction of tetracycline concentrations with generations (*F* = 2.65; *df* = 12, 621; *p* < 0.05) (Table 2). The average survival durations were 10.08 ± 1.12–12.8 ± 0.81, 3.03 ± 0.39–8.10 ± 1.17, 6.67 ± 1.07–14.54 ± 0.58, and 1.29 ± 0.08–8.33 ± 1.16 days for the control and tetracycline LC_5_-, LC_15_-, and LC_35_-fed females, respectively. The average longevity decreased significantly with the increase in tetracycline concentration. When treated with tetracycline LC_5_, the longevity of *T. pretiosum* females decreased significantly during the first six generations (*p* < 0.05), and there were no significant differences found during the following four generations compared with the control. Feeding *T. pretiosum* females with tetracycline LC_15_ and LC_35_ significantly shortened their longevity over the ten generations (all *p* < 0.05) (Figure 2a). For parasitism, both tetracycline concentrations (*F* = 117.65; *df* = 3, 621; *p* < 0.001) and generations (*F* = 28.27; *df* = 4, 621; *p* < 0.001) significantly impacted the average number of eggs parasitized by *T. pretiosum*, as did the interaction (*F* = 3.24; *df* = 12, 621; *p* < 0.001) (Table 2). The average numbers of eggs parasitized by *T. pretiosum* were 100.8 ± 10.09–122.8 ± 10.36, 33.00 ± 7.54–138.2 ± 7.69, 5.34 ± 2.58–55.13 ± 9.21, and 0.58 ± 0.34–59.13 ± 11.2 per female for the control and tetracycline LC_5_, LC_15_, and LC_35_ treatments, respectively. When feeding *T. pretiosum* females with tetracycline LC_5_, the average number of parasitized eggs decreased significantly during the first six generations compared with that of females fed with only honey–water (*p* < 0.05). In addition, parasitized eggs significantly reduced when treated with tetracycline LC_15_ and LC_35_ during the ten generations (all *p* < 0.05) (Figure 2b). Similar to parasitism, both tetracycline concentrations (*F* = 107.65; *df* = 3, 621; *p* < 0.001) and generations (*F* = 27.97; *df* = 4, 621; *p* < 0.001) significantly impacted the fecundity of *T. pretiosum*, and the interaction of tetracycline concentrations with generations also significantly impacted the fecundity (*F* = 3.06; *df* = 12, 621; *p* < 0.001) (Table 2). A significant reduction in fecundity was also observed when the egg-laying females of *T. pretiosum* were exposed to various tetracycline treatments. *T. pretiosum* females in the control group produced more offspring than those in the tetracycline LC_5_ treatment group, which only occurred during the first six generations (*p* < 0.05) and not in the following four generations. With an increase in the tetracycline concentration, *T. pretiosum* produced less offspring over the ten generations. The number of *T. pretiosum* progeny per female ranged from 5.44 ± 2.45 to 52.13 ± 9.05 and from 1.58 ± 0.84 to 56.4 ± 10.51 for tetracycline LC_15_ and LC_35_ treatments, respectively, and these values were significantly lower than those of the control (from 90.53 ± 8.41 to 110.80 ± 9.27 per female) (all *p* < 0.05) (Figure 2c).

Females exposed to 25% honey–water produced only females during the first three days, and they subsequently produced very few males and intersex offspring. However, females exposed to antibiotic treatments (tetracycline LC_5_, LC_15_, and LC_35_) produced male and intersex offspring during the first two days, and they subsequently produced more males and intersex offspring. Both tetracycline concentrations (*F* = 131.05; *df* = 3, 401; *p* < 0.001) and generations (*F* = 12.80; *df* = 4, 401; *p* < 0.001) significantly impacted the proportion of female offspring, as did the interaction (*F* = 2.17; *df* = 12, 401; *p* < 0.05) (Table 2). After being fed tetracycline, the proportion of female offspring significantly decreased in the control as the generations increased (Figure 2d). In contrast, the proportion of male offspring in the tetracycline treatments was significantly higher than that in the control over the ten generations (all *p* < 0.05), but there was no significant difference in the male ratio among the tetracycline treatments under the tested generation (Figure 2e). In addition, the tetracycline concentrations (*F* = 121.41; *df* = 3, 401; *p* < 0.001), generations (*F* = 15.15; *df* = 4, 401; *p* < 0.001), and the interaction of tetracycline concentrations with generations (*F* = 2.72; *df* = 12, 401; *p* < 0.05) significantly impacted the proportion of male offspring (Table 2). As for the intersex percentage of offspring, there was no significant difference among the tetracycline treatments over the ten generations compared with the control. The intersex percentages were 1.94 ± 0.65–2.76 ± 0.55%, 1.76 ± 0.23–5.93 ± 1.60%, 1.04 ± 0.71–8.02 ± 2.70%, and 0.87 ± 0.65–5.05 ± 3.80% for the control and tetracycline LC_5_, LC_15_ and LC_35_ treatments, respectively (Figure 2f).

### 3.3. Sublethal Effects of Tetracycline on Life Table Parameters of T. pretiosum

The life table parameters of *T. pretiosum* treated with sublethal concentrations of tetracycline were calculated, and the results are shown in Table 3. Under successive stress treatments of sublethal concentrations of tetracycline, the net reproduction rate (*R*_0_), intrinsic rate of natural increase (*r_m_*), finite rate of increase (*λ*), mean generation time (*T*), and doubling time (*DT*) of *T. pretiosum* were significantly lower than those in the control over the ten generations. The higher the concentration of tetracycline, the smaller the population parameter in each generation. In addition, the life table parameters of *T. pretiosum* slightly increased as the number of treated generations increased, indicating that the adaptability of *T. pretiosum* to tetracycline gradually increased.

### 3.4. Sublethal Effects of Tetracycline on Wolbachia Titer in T. pretiosum

The *Wolbachia* titers in *T. pretiosum* treated with sublethal concentrations of tetracycline were determined by assessing the copy number of the *Wsp* gene using AQ-PCR. The results are shown in Table 4. After treatment with tetracycline LC_5_, LC_15_, and LC_35__,_ the copy numbers of *Wsp* in females were significantly lower than those in the control during the first two generations. The *Wolbachia* titer was still at a lower level after females fed tetracycline LC_5_ for ten generations. However, after exposure to tetracycline LC_15_ and LC_35_, *Wolbachia* titers showed varying degrees of ups and downs as the treatment times increased.

## 4. Discussion

In our current study, the *Wolbachia* titer in *T. pretiosum* sharply reduced after two generations of antibiotic treatments. However, it could be detected at a lower dose even after ten successive generations of antibiotic treatments, indicating that tetracycline did not completely remove *Wolbachia* from *T. pretiosum*. In a previous study, *Wolbachia* in arthropods was removed by antibiotics at different concentrations and time durations [30]. Similar results were reported by Wang et al. [26], Zhong and Li [31], and Zha et al. [32]. However, several studies have shown that *Wolbachia* can be successfully eliminated by antibiotic treatments. It is also possible for tetracycline treatment success to vary depending on the *Wolbachia* strain—some strains appear to be more resistant than others. For example, tetracycline has been used to remove the *Wolbachia* endosymbionts from *Litomosoides sigmodontis* [33]. Li et al. [27] monitored the population dynamics of *Wolbachia* in *Laodelphax striatellus* under successive stress treatments of tetracycline, and they found that *Wolbachia* density in females and males decreased to 0 after using tetracycline treatment over two generations. Pike and Kingcombe [34] found that *Wolbachia* could be successfully eliminated from *Folsomia candida* using rifampicin over two generations. Grenier et al. [35] showed that *Wolbachia* was not recovered in *Trichogramma* at the 14th generation after the 1st generation was treated with tetracycline treatment.

In addition, different types of antibiotics have different effects. The mode of action of tetracycline is to inhibit protein synthesis by preventing the association of aminoacyl-tRNA with the bacterial ribosome [36,37]. In contrast, the mode of action of rifampicin is to interfere with nucleic acid synthesis by binding to prokaryotic DNA-dependent RNA polymerase, as well as inhibiting the transcription of DNA to message RNA [33,38]. These different modes of action could explain why different antibiotics produce different effects. For example, rifampicin was found to be effective in eliminating *Wolbachia* from *Folsomia candida**,* whereas tetracycline at the same concentration was not [34]. Previous studies showed that *Wolbachia* could be removed by feeding insects sugar water containing antibiotics, but the complete elimination of *Wolbachia* required at least three to six successive generations of antibiotic treatment [39,40,41]. Liu et al. [42] described a novel approach to eliminate *Wolbachia* from *Nasonia vitripennis* by feeding pupae with rifampicin to two generations. Therefore, different antibiotics and methods could be used to try to eliminate *Wolbachia* from *T. pretiosum* in further research.

In Figure 1, the line treated with tetracycline LC_5_ developed a greater longevity and fecundity by F_10_ despite having very low *Wolbachia* titers. Unfortunately, because there was no *Wolbachia*-free line in the experiment, we were unable to determine the direct effects of tetracycline on fitness and the effects of *Wolbachia* removal. However, after ten generations of antibiotic treatment, we successfully obtained the bisexual recovery strain of *T. pretiosum* through backcrossing females from an isofemale line with males for several generations. Although the female adults of this bisexual strain had recovered to produce female and male offspring normally, *Wolbachia* could still be detected at a very low concentration using AQ-PCR (unpublished data), implying that *Wolbachia* could have evolved to be essential for their hosts. Moreover, the dose-dependent effects of tetracycline on longevity and fecundity (but not on the *Wolbachia* titer or sex ratio) suggested that tetracycline treatment had substantial sublethal effects on fitness, but it was unclear what concentration was required to have an effect on *Wolbachia* itself. The changes in longevity and fecundity across generations also suggested a selection response to the sublethal doses of tetracycline. In addition, *Wolbachia* titers showed varying degrees of increases and decreases in progeny after exposure to tetracycline LC_15_ and LC_35_ as treatment times increased. It might be that different individuals developed different levels of resistance to tetracycline. Due to the long-term co-adaptation between *Wolbachia* and their native hosts, the removal of *Wolbachia* could have adverse effects on the hosts.

We also found that *Wolbachia* affected sex determination in *T. pretiosum*. The sex ratio of offspring became more male-biased as the *Wolbachia* titer decreased. The main factor of sex determination in Hymenoptera is haplodiploidy. Unfertilized eggs develop into haploid males, and fertilized eggs develop into females. Another common sex-determination model in Hymenoptera is complementary sex determination (CSD). Heterozygous diploid eggs develop into females, and homozygous diploid eggs develop into males, but the males are inviable or sterile [6]. Ning et al. [25] speculated that the sex determination mechanisms of *Trichogramma* wasps might conform to genome imprinting. In this model, haploid eggs only contain a maternally imprinted sex-determination gene, and then they develop into males. Diploid eggs contain a maternally imprinted sex-determination gene and a paternally derived nonimprinted copy, and then they develop into diploid females. In addition, genomic imprinting is usually caused by DNA methylation in organisms [43]. With regard to *T. pre^W+^* in this study, we assumed that genome imprinting was retained in the offspring, and *Wolbachia* inhibited the effects of genome imprinting. When tetracycline treatments reduced the *Wolbachia* titer, the effect of parental generation genome imprinting became stronger, and more male offspring were produced. Furthermore, the genomic conflict between *Wolbachia* and the host’s nuclear genome has been proposed to explain the influence of *Wolbachia* on sex determination in *Trichogramma*. Because *Wolbachia* is cytoplasmically inherited, cytoplasmic genes are only transmitted through females and result in a female bias, while nuclear genes favor a balanced sex ratio by inhibiting the action of cytoplasmic elements [44]. In this case, *Wolbachia* would try to increase its transmission level, whereas the nuclear genes of hosts would try to suppress *Wolbachia*. In addition, numerous studies have shown that high temperatures decrease *Wolbachia* titers in *Trichogramma* wasps and produce more males and intersexes, resulting in a change in the original reproduction method and the sex ratio of offspring [25,45]. The sex determination in thelytokous reproduction associated with bacteria is more complicated, and the mechanism is not fully understood at present [3,46,47].

In conclusion, we used AQ-PCR to detect the *Wolbachia* titer, and we discovered that tetracycline treatments were able to partially remove *Wolbachia* from *T. pretiosum*. Our results clearly show that sublethal concentrations of tetracycline have adverse effects on the development of *T. pretiosum* and that the *Wolbachia* titer affects the sexual development of *T. pretiosum* eggs.

## Figures and Tables

**Figure 1 insects-13-00559-f001:**
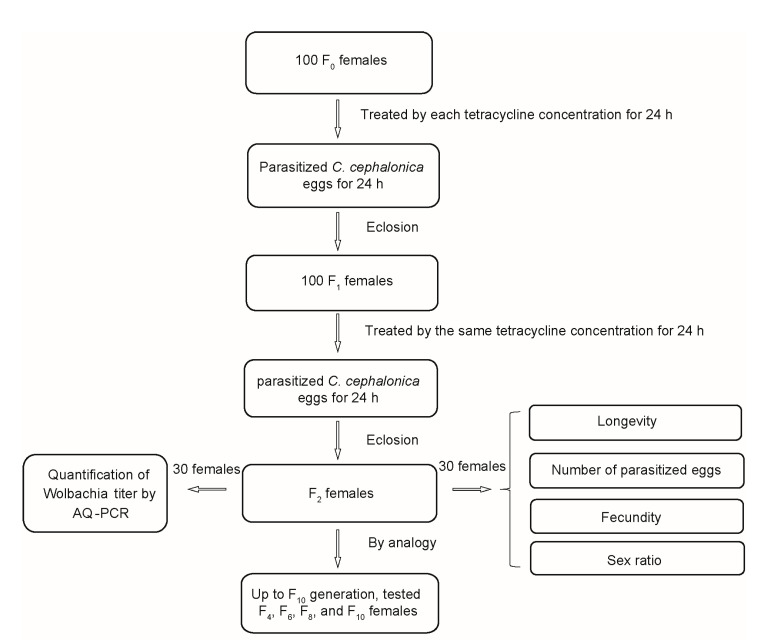
A flow chart of tetracycline treatment for *T. pretiosum*.

**Figure 2 insects-13-00559-f002:**
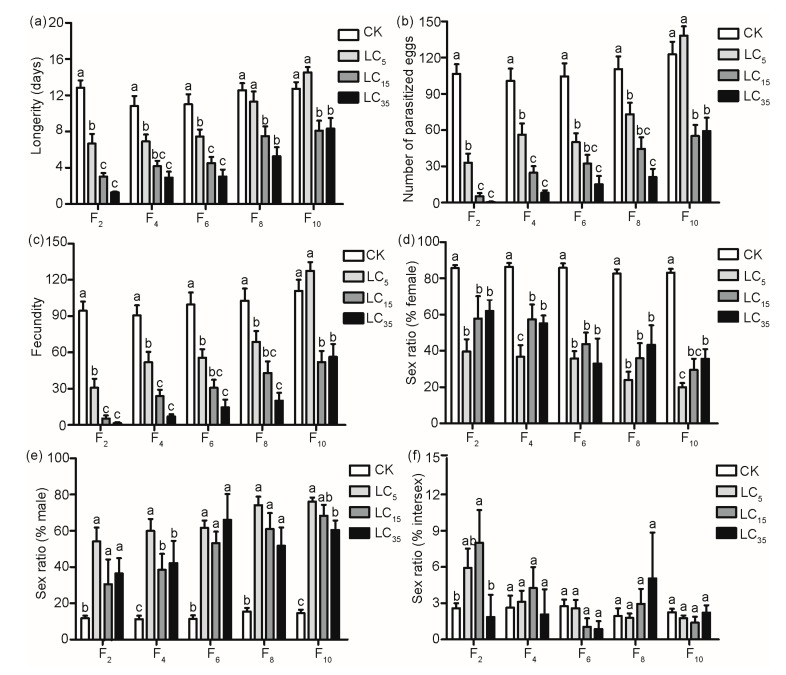
The effects of sublethal concentrations of tetracycline on development and reproduction of *T. pretiosum.* (**a**) Longevity, (**b**) number of parasitized eggs, (**c**) fecundity, (**d**) sex ratio (% female), (**e**) sex ratio (% male), and (**f**) sex ratio (% intersex) of *T. pretiosum* treatment with sublethal concentrations (LC_5_, LC1_5_, and LC_35_) of tetracycline. CK: non-tetracycline. The values are means ± SEM; means with different letters are significantly different (*p* < 0.05).

**Table 1 insects-13-00559-t001:** Toxicity of tetracycline to *T. pretiosum* females.

Antibiotic	Slope	LC_5_ (mg mL^−1^) (95%CI)	LC_15_ (mg mL^−1^) (95%CI)	LC_35_ (mg mL^−1^) (95%CI)
Tetracycline	2.56 (0.36)	0.79 (0.36–1.21)	1.37 (0.79–1.85)	2.46 (1.82–3.00)

**Table 2 insects-13-00559-t002:** Analysis of effects of tetracycline concentrations and generations on biological parameters of treated female parasitoids and their offspring.

Variable	Source of Variation	*df*	*F*	*p*-Value
Longevity	Tetracycline concentration (A)	3	84.2	<0.001
	Generation (B)	4	25.57	<0.001
	(A) × (B)	12	2.65	0.002
Number of parasitized eggs	Tetracycline concentration (A)	3	117.65	<0.001
	Generation (B)	4	28.27	<0.001
	(A) × (B)	12	3.24	<0.001
Fecundity	Tetracycline concentration (A)	3	107.65	<0.001
	Generation (B)	4	27.97	<0.001
	(A) × (B)	12	3.06	<0.001
Sex ratio (% female)	Tetracycline concentration (A)	3	131.05	<0.001
	Generation (B)	4	12.8	<0.001
	(A) × (B)	12	2.17	0.012
Sex ratio (% male)	Tetracycline concentration (A)	3	121.41	<0.001
	Generation (B)	4	15.15	<0.001
	(A) × (B)	12	2.72	0.001
Sex ratio (% intersex)	Tetracycline concentration (A)	3	1.1	0.347
	Generation (B)	4	2.7	0.03
	(A) × (B)	12	1.76	0.053

**Table 3 insects-13-00559-t003:** Life table parameters of *T. pretiosum* treatment with sublethal concentrations of tetracycline for ten generations.

Generation	Tetracycline Concentration	*R* _0_	*r_m_*	*λ*	*T* (d)	*DT*
F_2_	CK	76.48	0.29	1.34	14.77	2.37
	LC_5_	7.55	0.18	1.20	11.51	3.83
	LC_15_	2.75	0.09	1.09	11.50	7.67
	LC_35_	2.40	0.08	1.08	11.56	8.63
F_4_	CK	70.61	0.30	1.35	14.39	2.30
	LC_5_	13.73	0.23	1.26	11.50	3.00
	LC_15_	10.27	0.20	1.23	11.50	3.45
	LC_35_	5.17	0.15	1.15	11.62	4.60
F_6_	CK	82.94	0.32	1.37	13.90	2.16
	LC_5_	16.04	0.24	1.27	11.77	2.88
	LC_15_	12.07	0.20	1.22	12.36	3.45
	LC_35_	3.66	0.11	1.12	11.54	6.27
F_8_	CK	84.15	0.30	1.34	14.98	2.30
	LC_5_	11.69	0.21	1.23	11.90	3.29
	LC_15_	8.65	0.18	1.19	12.25	3.83
	LC_35_	4.60	0.13	1.14	11.55	5.30
F_10_	CK	87.83	0.31	1.36	14.56	2.23
	LC_5_	23.11	0.26	1.30	12.05	2.23
	LC_15_	10.09	0.19	1.21	11.96	3.63
	LC_35_	15.70	0.21	1.23	13.05	3.29

*R*_0_: net reproduction rate; *r_m_*: intrinsic rate of natural increase; *λ*: finite rate of increase; *T*: mean generation time; *DT*: doubling time.

**Table 4 insects-13-00559-t004:** Dynamics of *Wolbachia* titer in *T. pretiosum* treated with sublethal concentrations of tetracycline for ten generations.

Generation	Tetracycline Concentration	Copies μL^−1^	Significance
F_2_	CK	10,241.97 ± 373.91	a
	LC_5_	259.51 ± 80.29	b
	LC_15_	74.55 ± 7.31	b
	LC_35_	32.75 ± 1.36	b
F_4_	CK	10,005.82 ± 298.34	a
	LC_5_	89.78 ± 20.81	b
	LC_15_	387.12 ± 26.01	b
	LC_35_	111.22 ± 9.20	b
F_6_	CK	11,003.97 ± 425.56	a
	LC_5_	71.12 ± 0.87	b
	LC_15_	242.76 ± 20.71	b
	LC_35_	91.52 ± 2.84	b
F_8_	CK	10,653.92 ± 303.91	a
	LC_5_	28.77 ± 3.56	c
	LC_15_	75.4 ± 2.82	c
	LC_35_	686.84 ± 33.86	b
F_10_	CK	10,846.11 ± 429.88	a
	LC_5_	83.75 ± 7.20	b
	LC_15_	42.44 ± 1.45	b
	LC_35_	469.23 ± 57.71	b

The values are means ± SEM; means with different letters are significantly different (*p* < 0.05).

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
