# Peer review of "Effects of Sublethal Concentrations of Tetracycline Hydrochloride on the Biological Characteristics and Wolbachia Titer in Parthenogenesis Trichogramma pretiosum"

_insects, 2022, doi:10.3390/insects13060559_

Round 1

Reviewer 1 Report

Dear,

The manuscript "Effects of sublethal concentrations of tetracycline hydrochloride on the biological characteristics and Wolbachia titer in parthenogenesis Trichogramma pretiosum" which aims to "The objectives of this study were to investigate: 99
(1) whether Wolbachia affected biological characteristics of T. preW+, (2) whether Wolbachia could be removed by long-term tetracycline treatments to obtain uninfected line, (3) and to explore the interaction of Wolbachia titer and parthenogenesis of T. preW+. The research aimed at providing a new insight for understanding of the complex interaction between Wolbachia and its host" has scientific merit and deserves to be published in the Journal Insects. The manuscript presents imputations to the subject of the effect that tetracycline has on the endosymbionts of Trichogramma pretiosum.

Regards,

Author Response

Response: Thank you for your letter and the comments concerning our manuscript entitled “Effects of sublethal concentrations of tetracycline hydrochloride on the biological characteristics and Wolbachia titer in parthenogenesis Trichogramma pretiosum” (Manuscript ID: insects-1718396). Those comments are all valuable and very helpful for improving our paper. We made corrections wherever appropriate and answered the comments.

Reviewer 2 Report

 I appreciate the big amount of work done and the scope of the research, which addresses the effects of tetracycline on the development of Trichogramma pretiosum and population dynamic of symbiotic Wolbachia in it. Although, I think that the paper has been well organized and well detailed, I have the followings to improve paper's quality:

Introduction

The Introduction lacks the general information on how many species of Trichogramma genus have been tested for natural Wolbachia infection. What is the frequency of infection with Wolbachia in Trichogramma wasps?

L45 Arrhenotoky, known as arrhenotokous parthenogenesis, is also a form of parthenogenesis. So it should not be opposed to " Parthenogenesis or thelytoky is another mode of reproduction.."

L48-50  In text: "This form of reproduction in Trichogramma is often infected with Wolbachia bacteria harbored in its eggs, where Wolbachia can change chromosome behaviors of their hosts by gamete duplication during the first mitotic nuclear division." - The form of reproduction cannot be infected with Wolbachia. Should be rephrase.

Materials and Methods, Results

L 187 "The target genes from each symbiont were amplified, and the obtained PCR products were purified. "

It was only one gene studied - wsp

L191, 200 - 100 ug mL-1  -  µg/mL

L205  " 6.023 × 1023 Number of molecules in 1 mol (Avogadro constant)"

But, Avogadro's number is an absolute number: there are 6.022×1023 elementary entities in 1 mole. This will not fundamentally change anything in the calculation of the results, but in a scientific article one must be accurate.

L218, 240 "sex  radio"  should be changed to   "sex ratio"

Table 3 The life-table parameters of T. pretiosum R0, rm, λ, T, and DT should be spelled out  in the footnotes to the table and in the text.

Table 4   Significance of a,b,c  should be explained in the footnotes.

Discussion

It is necessary to discuss why Wolbachia titers showed varying degrees of increase and decrease in progeny after exposure to tetracycline LC15 and LC35 as treatment time increased.

There are many published studies on the effect of temperature on Wolbachia and, subsequently, on sex determination in Trichogramma wasps. Your results can be discussed in connection with this.

Regarding the influence of Wolbachia on sex determination in Trichogramma, not only assumed genome imprinting, but other possible genetic conflicts between host genes and Wolbachia should be discussed.

L356 Amend the misspelling wolbachia to Wolbachia

Author Response

I appreciate the big amount of work done and the scope of the research, which addresses the effects of tetracycline on the development of Trichogramma pretiosum and population dynamic of symbiotic Wolbachia in it. Although, I think that the paper has been well organized and well detailed, I have the followings to improve paper's quality:

Introduction

The Introduction lacks the general information on how many species of Trichogramma genus have been tested for natural Wolbachia infection. What is the frequency of infection with Wolbachia in Trichogramma wasps?

Response: Thanks very much for your great suggestion. We have added them. However, the frequency of infection with Wolbachia in Trichogramma wasps is low, and there are 16 thelytokous Trichogramma species induced by a natural Wolbachia infection. 

L45 Arrhenotoky, known as arrhenotokous parthenogenesis, is also a form of parthenogenesis. So it should not be opposed to " Parthenogenesis or thelytoky is another mode of reproduction.."

Response: Thanks very much for your great suggestion. We have rewritten these sentences.

L48-50  In text: "This form of reproduction in Trichogramma is often infected with Wolbachia bacteria harbored in its eggs, where Wolbachia can change chromosome behaviors of their hosts by gamete duplication during the first mitotic nuclear division." - The form of reproduction cannot be infected with Wolbachia. Should be rephrase.

Response: Thanks very much for your great suggestion. We have revised it.

Materials and Methods, Results

L 187 "The target genes from each symbiont were amplified, and the obtained PCR products were purified. "

It was only one gene studied - wsp

Response: Thanks very much for your great suggestion. We have revised them.

L191, 200 - 100 ug mL-1  -  µg/mL

Response: Thanks very much for your great suggestion. We have revised it.

L205  " 6.023 × 1023 Number of molecules in 1 mol (Avogadro constant)"

But, Avogadro's number is an absolute number: there are 6.022×1023 elementary entities in 1 mole. This will not fundamentally change anything in the calculation of the results, but in a scientific article one must be accurate.

Response: Thanks very much for your great suggestion. We have revised it.

L218, 240 "sex  radio"  should be changed to   "sex ratio"

Response: Thanks very much for your great suggestion. We have revised them.

Table 3 The life-table parameters of T. pretiosum R0, rm, λ, T, and DT should be spelled out. 

Response: Thanks very much for your great suggestion. We have added them in the footnotes to the table and in the text.

Table 4   Significance of a,b,c  should be explained in the footnotes.

Response: Thanks very much for your great suggestion. We have added them.

Discussion

It is necessary to discuss why Wolbachia titers showed varying degrees of increase and decrease in progeny after exposure to tetracycline LC15 and LC35 as treatment time increased.

Response: Thanks very much for your great suggestion. We have added them.

There are many published studies on the effect of temperature on Wolbachia and, subsequently, on sex determination in Trichogramma wasps. Your results can be discussed in connection with this.

Response: Thanks very much for your great suggestion. We have added them.

Regarding the influence of Wolbachia on sex determination in Trichogramma, not only assumed genome imprinting, but other possible genetic conflicts between host genes and Wolbachia should be discussed.

Response: Thanks very much for your great suggestion. We have added them.

L356 Amend the misspelling wolbachia to Wolbachia

Response: Thanks very much for your great suggestion. We have revised it.

Reviewer 3 Report

General comments

In this study, the authors investigate cross-generational effects of sublethal concentrations of tetracycline on Wolbachia and life history traits in Trichogramma wasps. They find dramatic effects of tetracycline treatment on several traits, including longevity, fecundity and sex ratio. Tetracycline treatment also dramatically reduced Wolbachia titers but complete removal was not achieved even at the highest concentration and with repeated exposure across generations. There were also interesting dose-dependent and generational effects, where higher concentrations of tetracycline induced greater fitness costs but did not increase the proportion of males in a dose-dependent manner. Furthermore, there were dramatic changes in the treated lines across generations, where longevity and fecundity increased to the point where they were higher in some treated lines compared to the untreated line. These effects highlight complex interactions between tetracycline treatment and life history parameters in this species.

Unfortunately, because there is no Wolbachia-free line in the experiment, the authors are unable to separate direct effects of tetracycline on fitness and the effects of Wolbachia removal. The dose-dependent effects of tetracycline on longevity and fecundity (but not on Wolbachia titer or sex ratio)  suggest that tetracycline treatment has substantial sublethal effects on fitness, but it is unclear what concentration is required to have an effect on Wolbachia itself. The changes in longevity and fecundity across generations also suggest a selection response to the sublethal doses of tetracycline. The authors do not discuss these observations or limitations in detail- most of the discussion focuses on summarizing previous studies, however direct comparisons with other studies are difficult because the effects of tetracycline differ dramatically depending on the Wolbachia strain, host species and method of treatment.

Lines 87-91 – These studies of Wolbachia in mosquitoes are not relevant to the current paper

Table 4 – The Wolbachia copy number for each control (CK) is identical across generations which is biologically implausible and appears to be an error. The table legend should also indicate what the errors are for each value (standard errors?)

Line 346 – It is also possible for tetracycline treatment success to vary depending on the Wolbachia strain- some strains appear to be more resistant than others

Line 356 – This is an incorrect generalization that is highly dependent on the insect species, concentration used and method of exposure (feeding adults versus feeding larvae etc) – some studies have successfully eliminated Wolbachia in a single generation.

Line 364 – It is unclear from this study if Wolbachia infection is advantageous to reproduction. In Figure 1, the line treated with LC5 of tetracycline eventually had greater longevity and fecundity by F10 despite having very low Wolbachia titers. Without including a Wolbachia-free line in the experiment it is unclear if the fitness changes across time are due to changes in Wolbachia titer alone or other effects of tetracycline on fitness.

Author Response

General comments
In this study, the authors investigate cross-generational effects of sublethal concentrations of tetracycline on Wolbachia and life history traits in Trichogramma wasps. They find dramatic effects of tetracycline treatment on several traits, including longevity, fecundity and sex ratio. Tetracycline treatment also dramatically reduced Wolbachia titers but complete removal was not achieved even at the highest concentration and with repeated exposure across generations. There were also interesting dose-dependent and generational effects, where higher concentrations of tetracycline induced greater fitness costs but did not increase the proportion of males in a dose-dependent manner. Furthermore, there were dramatic changes in the treated lines across generations, where longevity and fecundity increased to the point where they were higher in some treated lines compared to the untreated line. These effects highlight complex interactions between tetracycline treatment and life history parameters in this species.
Unfortunately, because there is no Wolbachia-free line in the experiment, the authors are unable to separate direct effects of tetracycline on fitness and the effects of Wolbachia removal. The dose-dependent effects of tetracycline on longevity and fecundity (but not on Wolbachia titer or sex ratio)  suggest that tetracycline treatment has substantial sublethal effects on fitness, but it is unclear what concentration is required to have an effect on Wolbachia itself. The changes in longevity and fecundity across generations also suggest a selection response to the sublethal doses of tetracycline. The authors do not discuss these observations or limitations in detail- most of the discussion focuses on summarizing previous studies, however direct comparisons with other studies are difficult because the effects of tetracycline differ dramatically depending on the Wolbachia strain, host species and method of treatment.
Lines 87-91 – These studies of Wolbachia in mosquitoes are not relevant to the current paper

Response: Thanks very much for your great suggestion. We have deleted them.

Table 4 – The Wolbachia copy number for each control (CK) is identical across generations which is biologically implausible and appears to be an error. The table legend should also indicate what the errors are for each value (standard errors?)

Response: We are so sorry for the error about the Wolbachia copy number for each control. We had tested the samples again and revised them. In addition, we had added the information about the values in the footnotes to the table.  

Line 346 – It is also possible for tetracycline treatment success to vary depending on the Wolbachia strain- some strains appear to be more resistant than others

Response: Thanks very much for your great suggestion. We have added them.

Line 356 – This is an incorrect generalization that is highly dependent on the insect species, concentration used and method of exposure (feeding adults versus feeding larvae etc) – some studies have successfully eliminated Wolbachia in a single generation.

Response: Thanks very much for your great suggestion. We have revised it.

Line 364 – It is unclear from this study if Wolbachia infection is advantageous to reproduction. In Figure 1, the line treated with LC5 of tetracycline eventually had greater longevity and fecundity by F10 despite having very low Wolbachia titers. Without including a Wolbachia-free line in the experiment it is unclear if the fitness changes across time are due to changes in Wolbachia titer alone or other effects of tetracycline on fitness.

Response: Thanks very much for your great suggestion. We have revised it.

Round 2

Reviewer 3 Report

I thank the authors for considering my comments and making several improvements to the manuscript. However, the authors have not addressed my key concerns:

1. Because there is no Wolbachia-free line in the experiment, the authors are unable to separate direct effects of tetracycline on fitness and the effects of Wolbachia removal on fitness. This limitation is still not addressed in the revised manuscript. The authors need to revise their conclusion that Wolbachia is advantageous to reproduction, because the experimental design does not allow them to conclude whether changes in fitness are due to the negative effects of tetracycline on fitness or Wolbachia removal.

2. The authors still do not provide any explanation for the dose-dependent and cross-generational effects. For instance, the changes in longevity and fecundity across generations suggest a selection response to the sublethal doses of tetracycline.

Unfortunately, there are also few cases where the authors indicate that they have revised the manuscript in response to a comment, but there are no apparent changes to the manuscript in the relevant section. See below for two examples where the relevant sections remain unchanged.

Original reviewer comment: Lines 87-91 – These studies of Wolbachia in mosquitoes are not relevant to the current paper

Response: Thanks very much for your great suggestion. We have deleted them.

Line 356 – This is an incorrect generalization that is highly dependent on the insect species, concentration used and method of exposure (feeding adults versus feeding larvae etc) – some studies have successfully eliminated Wolbachia in a single generation.

Response: Thanks very much for your great suggestion. We have revised it.

Author Response

Dear Reviewer:

Thank you for your commens concerning our manuscript entitled “Effects of sublethal concentrations of tetracycline hydrochloride on the biological characteristics and Wolbachia titer in parthenogenesis Trichogramma pretiosum” (Manuscript ID: insects-1718396). Those comments are all valuable and very helpful for improving our paper. We made corrections wherever appropriate and answered the comments.

Sincerely,

Xiaoge Nian

I thank the authors for considering my comments and making several improvements to the manuscript. However, the authors have not addressed my key concerns:
1. Because there is no Wolbachia-free line in the experiment, the authors are unable to separate direct effects of tetracycline on fitness and the effects of Wolbachia removal on fitness. This limitation is still not addressed in the revised manuscript.

Response: Thanks very much for your great suggestion. We were so regret that we did not obtain Wolbachia-free line. So we were unable to separate direct effects of tetracycline on fitness and the effects of Wolbachia removal on fitness. We added your great suggestion in the discussion. However, after ten generations of antibiotics treatment, we successfully obtained the bisexual recovery strain of T. pretiosum through backcrossing females from an isofemale line with males for several generations. Although the female adults of this bisexual strain had recovered to produce female and male offspring normally, Wolbachia could still be detected at a very low concentration by AQ-PCR (unpublished data), which implying Wolbachia could had been evolved to be essential for their hosts.

The authors need to revise their conclusion that Wolbachia is advantageous to reproduction, because the experimental design does not allow them to conclude whether changes in fitness are due to the negative effects of tetracycline on fitness or Wolbachia removal.
Response: Thanks very much for your great suggestion. We have revised the conclusion throughout the manuscript.
2. The authors still do not provide any explanation for the dose-dependent and cross-generational effects. For instance, the changes in longevity and fecundity across generations suggest a selection response to the sublethal doses of tetracycline.
Response: Thanks very much for your great suggestion. We have added them in the discussion.

Original reviewer comment: Lines 87-91 – These studies of Wolbachia in mosquitoes are not relevant to the current paper

Response: Thanks very much for your great suggestion. We have deleted them.
Line 356 – This is an incorrect generalization that is highly dependent on the insect species, concentration used and method of exposure (feeding adults versus feeding larvae etc) – some studies have successfully eliminated Wolbachia in a single generation.

Response: Thanks very much for your great suggestion. We have deleted it.

Table 4 – The Wolbachia copy number for each control (CK) is identical across generations which is biologically implausible and appears to be an error. The table legend should also indicate what the errors are for each value (standard errors?)

Response: We are so sorry for the error about the Wolbachia copy number for each control. We had tested the samples again and revised them. In addition, we had added the information about the values in the footnotes to the table.  

Line 346 – It is also possible for tetracycline treatment success to vary depending on the Wolbachia strain- some strains appear to be more resistant than others

Response: Thanks very much for your great suggestion. We have added them.

Line 364 – It is unclear from this study if Wolbachia infection is advantageous to reproduction. In Figure 1, the line treated with LC5 of tetracycline eventually had greater longevity and fecundity by F10 despite having very low Wolbachia titers. Without including a Wolbachia-free line in the experiment it is unclear if the fitness changes across time are due to changes in Wolbachia titer alone or other effects of tetracycline on fitness.

Response: Thanks very much for your great suggestion. We have added it in the discussion.

Round 3

Reviewer 3 Report

I thank the authors for addressing my additional comments- I have no further suggestions for improvement of the scientific content, however some language editing is required.

This manuscript is a resubmission of an earlier submission. The following is a list of the peer review reports and author responses from that submission.